# Association between Sugar-Sweetened Beverage Consumption and Executive Function in Children

**DOI:** 10.3390/nu13124563

**Published:** 2021-12-20

**Authors:** Zhaohuan Gui, Shan Huang, Yican Chen, Yu Zhao, Nan Jiang, Shuxin Zhang, Yajun Chen

**Affiliations:** Department of Maternal and Child Health, School of Public Health, Sun Yat-sen University, Guangzhou 510080, China; guizhh@mail2.sysu.edu.cn (Z.G.); huangsh59@mail2.sysu.edu.cn (S.H.); chenyc55@mail2.sysu.edu.cn (Y.C.); zhaoy369@mail2.sysu.edu.cn (Y.Z.); jiangn63@mail2.sysu.edu.cn (N.J.); zhangshx26@mail2.sysu.edu.cn (S.Z.)

**Keywords:** children, executive function, sugar-sweetened beverages

## Abstract

The association between sugar-sweetened beverages (SSB) and executive function among children has been less investigated. We aimed to explore this topic. We randomly recruited 6387 children aged 6–12 years from five elementary schools in Guangzhou, China in 2019. Information on frequency and servings of children’s SSB consumption was assessed using a questionnaire. Children’s executive function was evaluated using parents’ ratings of the Behavioral Rating Inventory of Executive Function (BRIEF), which comprises eight subscales—including inhibit, shift, emotional control, initiate, working memory, plan/organize, organization of materials and monitor, as well as three composite indexes including behavioral regulation index (BRI), metacognition index (MI), and global executive index (GEC). SSB consumption was positively associated with all subscales and composite scores of BRIEF as well as higher risks of elevated executive difficulties, indicating poorer executive function. For example, children who drank SSB ≥2 times/week were related to higher scores of GEC (estimates, 95% confidence interval (CI): 2.44, 1.79 to 3.09) compared with those who never drank SSB. The odds ratio of elevated GEC associated with SSB consumption ≥2 times/week was 1.62 (95% CI: 1.34, 1.96) than non-consumers. The results of this study indicated that SSB consumption was associated with poorer executive function in children.

## 1. Introduction

The consumption of sugar-sweetened beverages (SSB) has been decreasing in most Western countries during the past two decades [1,2,3], but SSB is the major contributor of added sugar in the American diet and its consumption has been increasing worldwide [4,5,6]. The harmful impact of SSB on cardiometabolic health has been well documented, and it is associated with greater risks of obesity [7], hypertension [8], type 2 diabetes (independently of adiposity) [9], and cardiometabolic death [10]. An estimation based on globally representative data calculated that about 184,000 yearly deaths worldwide in 2010 were attributed to SSB consumption, and three quarters of this burden occurred in low- and middle-income countries [11]. This could be explained by the following several factors: (1) SSB has poor satiating property that can be consumed excessively [12]; (2) SSB displaces more nutritional foods and beverages [13]; and (3) SSB contain high levels of fructose [14,15], and fructose metabolism results in detrimental health outcomes at the organ and metabolic level [16].

In contrast to extensive research on the physical health impacts of SSB consumption, the association between SSB consumption and executive function has been less investigated. Executive function is an umbrella term involving a variety of interrelated, higher-level cognitive skills that were requisite for complex reasoning, goal-oriented activity, and self-regulatory behavior [17,18]. Although the windows of neurodevelopmental vulnerability occur during prenatal and early postnatal periods [19], high-order executive skills develop significantly from ages 6 to 10 which makes this period especially sensitive to perturbation [20]. Animal studies have shown that sugar could induce the increases in mediators of inflammation (e.g., IL-6 and IL-1β) and oxidative stress as well as decrease in neurotrophins, and these intermediate factors subsequently altered brain structure and function [21,22,23]. Therefore, it is reasonable to hypothesize that excessive consumption of SSB may harmfully impact the performance on executive function. To our knowledge, there were limited epidemiological studies investigating the association between SSB consumption and executive function in children [24,25,26,27], and results were inconsistent. In addition, the majority of these studies were implemented in high-income countries, and such studies from low- and middle-income countries such as China were scarce, where the level of SSB consumption was relatively low [5]. We therefore aimed to investigate the association between SSB consumption and executive function in children by using data from a cross-sectional study in Guangzhou, Southern China.

## 2. Materials and Methods

### 2.1. Study Design and Population

Data comes from a cross-sectional study, which was implemented in Guangzhou, southern China from April to May 2019. We performed a two-stage cluster sampling strategy to recruit study participants. First, we randomly selected five districts including three urban areas and two suburban areas in Guangzhou city. Second, we randomly selected one elementary school within each district, which generated five elementary schools. Finally, all students from the selected five schools were invited to participate. The above sampling strategy generated 8692 eligible participants, of whom 6883 children and parents agreed to participate, giving a response rate of 79.2%. We additionally excluded 496 children without information on SSB, executive function or possible important confounders, leaving a final sample of 6387 participants aged 6–12 years (Figure 1).

Approve for this study was granted by the Ethical Review Committee for Biomedical Research, Sun Yat-sen University. The study has been registered with China Clinical Trial Registry NCT03582709. Prior to data collection, all children and their parents/guardians gave the written informed consent.

### 2.2. Sugar-Sweetened Beverage Consumption

The following two questions, reported by children and their parents, were used to assess SSB consumption. Participants were asked about the frequency of SSB consumption from the question “In the past 7 days, how many times did your child drink SSB such as Coke, Sprite, Fruit drinks (Orange juice drink etc.), Nutrition Express, Red Bull?”. If they answered more than 0 times, they were further asked about the servings of SSB each time consumed from the question “On average, how many servings of SSB did your child drink each time? (One serving is equal to 250 mL)”. The distribution of SSB consumption status was highly skewed, and transformation of data was not feasible owing to the large number of people who reported never drinking SSB. Therefore, the frequency of SSB consumption was aggregated and then a new intake category was categorized in order to ensure an adequate number of participants in each group. SSB consumption was examined as a three-level variable: 0 time/week, 1 time/week, and ≥2 times/week. In addition, we calculated the total servings of SSB intake weekly (servings/week) using the product of times of SSB intake per week by the servings of SSB intake each time.

### 2.3. Executive Function

We used a parent-rated Behavioral Rating Inventory of Executive Function (BRIEF) to assess children’s executive function. The parental form of BRIEF provides an ecological assessment of executive function in everyday settings at home for children at 5 and 18 years of age [28]. It is valid, reliable [28], and widely used in epidemiological and clinical studies [29]. The BRIEF has been translated into Chinese and documented to have a high value of internal consistency, with Cronbach’s α from 0.70 to 0.96 [30]. The BRIEF comprises 86 items, which were grouped into eight subscales and three composite indexes. Scores of interest for this study were the behavioral regulation index (BRI), a sum of inhibit, shift, and emotional control subscales; the metacognition index (MI), a sum of initiate, working memory, plan/organize, organization of materials, and monitor subscales; and global executive composite (GEC), a sum of all eight subscale scores. The BRI, MI, and GEC were converted into T-scores (mean = 50, standard deviation (SD) = 10) and standardized by age and sex. Higher scores indicate more problems of executive function. Scores ≥60 (e.g., 1 SD from the mean) were classified as ‘elevated executive difficulties’.

### 2.4. Covariates

Via questionnaires, we obtained the following information on sociodemographic factors and lifestyles including age (years), sex (boys or girls), siblings (0 or ≥1), monthly household income (<5000 RMB, 5000–7999 RMB, 8000–11,999 RMB, ≥12,000 RMB or refused to answer), parental education (highest degree of each parent: below senior high, senior high, college, university or above), parental smoking status (never smokers if neither parent smoked, former or current smokers if either of parents were former or current smokers), outdoor exercise (<1 h/day, 1–1.9 h/day, 2–4 h/day, or >4 h/day), and dietary intakes (times/week). Information on dietary intakes was collected separately by 3 questions “In the past 7 days, how many times did your child eat (1) fried food; (2) fish or fish products; and (3) milk, or dairy-products?”. Body height and weight were measured with children lightly dressed and in bare feet, and body mass index (BMI) was calculated as weight in kilograms divided by height in meters.

### 2.5. Statistical Analyses

Differences in basic characteristics by SSB consumption groups were analyzed by using analysis of variance for continuous data and the chi-square test for categorical data. Data were presented as mean and SD for continuous variables or as number and percentage for categorical variables.

To examine the associations between SSB consumption and executive function, we performed general linear regression models for continuous data (BRIEF scores) as well as logistic regression models for binary data (elevated executive difficulties). The 0 time/week group was treated as the reference group, and model estimates were presented as regression coefficients (βs) or odds ratio (OR) and 95% confidence intervals (CIs). Three models of increasing covariates adjustment were conducted. With no adjustment in the unadjusted model, Model 1 included sex, age, only child, monthly household income, parental education, parental smoking status, outdoor exercise, and BMI as control variables. Model 2 included fried food, fish or fish products, and milk, or dairy-products as additional control variables. We performed trend tests by entering ordinal categorical variables as continuous variables in the three models. We then restricted to children who reported drinking SSB to assess the dose-response relationship between the total servings of SSB consumption a week (as a continuous variable) and executive function.

We performed subgroup analyses by sex (boys, girls), age (<10 years, ≥10 years), and BMI (normal weight, overweight/obesity) and also examined the effect modification by incorporating a multiplicative interaction term for SSB × modifier in the adjusted model.

Data analyses were conducted with SAS software (version 9.4, SAS Institute). All criteria for statistical significance were set at a two-tailed *p* < 0.05.

## 3. Results

### 3.1. Description of Study Participants

Among 6387 participants, the mean (SD) age was 8.6 (1.5) years, and 3410 (53.4%) were boys. A total of 4116 (64.4%) children reported consuming SSB, with 1918 (30.0%) having 1 time a week and 2198 (34.4%) having no less than 2 times a week (Table 1). The mean (SD) of servings of SSB intake a week among SSB consumers was 2.45 (2.57) (data not shown). Compared to non-consumers, children who consumed SSB were more likely to be older, boys, born to parents who smoked, and to have higher BMI, higher intakes of fried food, fish or fish products, and milk, or dairy-products (all *p* values < 0.05). The distribution of scores of BRIEF among SSB consumption is presented in Table A1.

### 3.2. Association between SSB Consumption and Executive Function

SSB consumption was significantly associated with inferior performance on executive function (Table 2, Table 3, and Table A2). In the adjusted model 2, children who drank SSB 1 time a week was positively related to all subscales and composite scores of BRIEF including inhibit, shift, emotional control, initiate, working memory, plan/organize, organization of materials, monitor, BRI, MI and GEC, with estimates ranging from 0.82 (95% CI: 0.18, 1.46) to 1.22 (95% CI: 0.59, 1.86), relative to participants who never drank SSB (*p*-values for all tests were <0.05) (Table 2). Similarly, children who drank SSB ≥2 times/week was positively associated with all subscales and composite scores of BRIEF, with estimates ranging from 1.55 (95% CI: 0.91, 2.19) to 2.50 (95% CI: 1.86, 3.14), compared to those children who were non-consumers (*p*-values for all trend tests <0.05). When only the children who had a habit of drinking SSB were analyzed, we found that each serving increase in SSB consumption was positively associated with all scores of BRIEF, with the estimates ranging from 0.21 (95% CI: 0.03, 0.40) to 0.35 (95% CI: 0.16, 0.54) (Table A2).

The odds ratios of elevated BRI and elevated MI in children who drank SSB 1 time a week were 1.23 (95% CI: 1.02, 1.50) and 1.21 (95% CI: 1.00, 1.47), respectively, compared to their counterparts who never drank SSB in the adjusted model 2 (*p* values for all tests were <0.05) (Table 3). Additionally, the odds ratios of elevated BRI, elevated MI, and elevated GEC in children who drank SSB ≥2 times a week were 1.45 (95% CI: 1.19, 1.76), 1.70 (95% CI: 1.41, 2.05) and 1.62 (95% CI:1.34, 1.96), respectively, compared to children who were non-consumers (*p*-values for all trend tests <0.05).

### 3.3. Effect Modification

We further assessed potential modification effects between SSB consumption with sex, age, and BMI on executive function. However, no significant modification effects were observed for sex, age, or BMI (Table A3).

## 4. Discussion

In our cross-sectional analysis of the large population-based sample, we observed that higher consumption of SSB was associated with poorer performance on executive function and high risk of executive dysfunction in children. The association between SSB consumption and executive function remained robust after adjustment for a wide range of covariates including sociodemographic factors, lifestyles, and diets.

In this large sample of nearly 6000 school children, about two-thirds of children reported consuming SSB, which was comparable with the study conducted among American children [31]. To our knowledge, there were a few studies examining the association between SSB consumption and executive function in children, and these studies yielded inconsistent observations [24,25,26,27]. For example, in agreement with the current results, a population-based cohort study of 1234 children in the US reported that each additional SSB serving consumed per day in early childhood was associated with a 2.4-point decrease in verbal intelligence evaluated by Kaufman Brief Intelligence Test at mid-childhood [25]. In another study conducted in Kuwait among 1370 adolescents aged 11–16 years, consumption of soft sugar drinks was inversely associated with non-verbal intelligence assessed by Raven’s Standard Progressive Matrices [24]. In contrast, an earlier meta-analysis that pooled 16 interventions found no overall association between sugar and glucose with cognition, although the control group in the concluded studies were administrated artificial sweeteners [27]. Similarly, in a recent study of 868 school children aged 8 to 10 years in the US, there was no significant association between SSB consumption and cognitive or academic outcomes including working memory, inhibitory control, mathematics, or English language arts score [26]. There were several differences between the described studies that could explain the inconsistent findings. These included the heterogeneity in study population (e.g., age, genetic background, and lifestyles), methods in quantifying SSB consumption as well as the instruments of executive function measurement. The findings of our study combining the studies mentioned above support the negative association between SSB consumption and executive function among children.

The biological mechanism underlying the relationship between SSB consumption and executive function are yet to be established. Increases in inflammation and oxidative stress, as well as decreases in neurotrophins, are the most plausible pathways proposed by previous studies [21,22,32]. Specifically, evidence from an animal study demonstrated that rats fed with sucrose-fructose drinks had increased mediators of inflammation in the dorsal hippocampus including IL-6 and IL-1β [21]. Rats exposed to a fructose-sweetened solution also displayed an increased level of oxidative stress and advanced glycation end products as well as decreased antioxidant enzymes in the frontal cortex [22]. Moreover, studies on rodent models have evidenced that fructose administration in rats was associated with a reduction in the hippocampal brain-derived neurotrophic factor, a protein that supports synaptic plasticity and circuit information [32]. These intermediators subsequently affect the executive function.

This study had several potential limitations. First, the cross-sectional study design cannot be used to infer the causality of SSB consumption and executive function, prospective cohort study or randomized controlled study in the future would be need to establish that level of causality inference. Second, recall bias and information bias from assessment of SSB consumption, parent-rated executive function, and questionnaire-based sociodemographic factors might be evitable. Third, SSB consumption was self-reported, and we did not objectively measure the concentrations of such exposure. Fourth, we did not measure other sources of sugar, and therefore intake of foods high in sugar could not be accounted for. Fifth, though a wide range of covariates were carefully adjusted in the model, residual confounding caused by unavailable data, such as parental mental health status, could exist.

## 5. Conclusions

In conclusion, the findings of this study found that SSB consumption was associated with poorer performance on executive function among children. Because excessive consumption of SSB is fairly common in many countries, the findings hold importance for informing policy makers to implement intervention strategies on reducing children’s access to SSB for promoting brain health.

## Figures and Tables

**Figure 1 nutrients-13-04563-f001:**
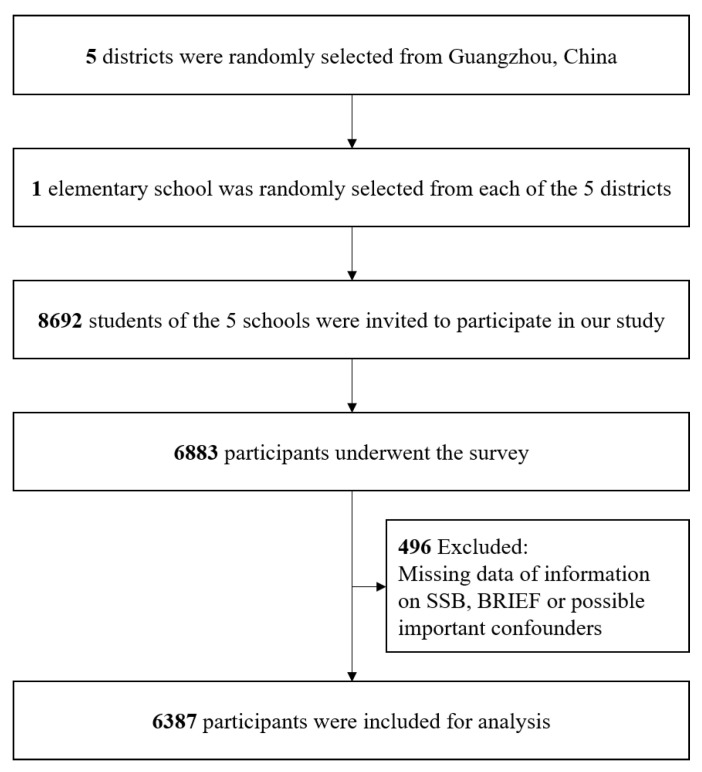
Flowchart of participant inclusion.

**Table 1 nutrients-13-04563-t001:** Study population characteristics according to SSB consumption.

Characteristics	Total Sample	SSB Consumption			*p* Value
		0 Time/Week	1 Time/Week	≥2 Times/Week	
*n*	6387	2271 (35.6)	1918 (30.0)	2198 (34.4)	
Age (years)	8.6 ± 1.5	8.4 ± 1.5	8.5 ± 1.5	8.9 ± 1.5	<0.0001
Sex					<0.0001
Boys	3410 (53.4)	1121 (49.4)	1019 (53.1)	1270 (57.8)	
Girls	2977 (46.6)	1150 (50.6)	899 (46.9)	928 (42.2)	
Siblings					0.090
0	2701 (42.4)	922 (40.8)	814 (42.5)	965 (44.0)	
≥1	3666 (57.6)	1338 (59.2)	1102 (57.5)	1226 (56.0)	
Monthly household income					0.053
<5000 RMB	848 (13.3)	298 (13.2)	259 (13.6)	291 (13.3)	
5000–7999 RMB	1366 (21.5)	478 (21.1)	410 (21.4)	478 (21.8)	
8000–11,999 RMB	1299 (20.4)	428 (18.9)	385 (21.0)	486 (22.2)	
≥12,000 RMB	2250 (35.3)	813 (36.0)	700 (36.5)	737 (33.6)	
Refused to answer	604 (9.5)	243 (10.8)	162 (8.5)	199 (9.1)	
Parental education					0.310
Below senior high school	181 (2.8)	60 (2.7)	57 (3.0)	64 (2.9)	
Senior high school	659 (10.4)	228 (10.1)	179 (9.3)	252 (11.5)	
College	1164 (18.3)	405 (17.9)	369 (19.3)	390 (17.8)	
University or above	4363 (68.5)	1567 (69.3)	1311 (68.4)	1485 (67.8)	
Parental smoking status					<0.0001
Never smokers	3716 (58.4)	1423 (38.3)	1116 (30.0)	1177 (31.7)	
Former smokers	749 (11.8)	233 (31.1)	224 (29.9)	292 (39.0)	
Current smokers	1902 (29.9)	604 (31.8)	576 (30.3)	722 (38.0)	
Outdoor exercise					0.975
<1 h/day	2743 (43.1)	965 (35.2)	838 (30.6)	940 (34.3)	
1–1.9 h/day	2807 (44.1)	1001 (35.7)	836 (29.8)	970 (34.6)	
2–4 h/day	609 (9.6)	215 (35.3)	180 (29.6)	214 (35.1)	
>4 h/day	208 (3.3)	144 (69.9)	39 (18.9)	23 (11.2)	
BMI (kg/m^2^)	16.8 ± 3.0	16.6 ± 3.0	16.7 ± 2.9	17.1 ± 3.1	<0.0001
Fried food (times/week)	0.8 ± 1.0	0.4 ± 0.8	0.7 ± 0.8	1.2 ± 1.2	<0.0001
Fish or fish products (times/week)	3.0 ± 2.1	2.9 ± 2.1	2.9 ± 2.0	3.1 ± 2.1	0.0001
Milk, or dairy-products (times/week)	6.5 ± 3.3	6.5 ± 7.0	6.3 ± 3.3	6.6 ± 3.2	0.014

Note: Descriptive statistics are presented as mean (standard deviation) and number (percentage) for continuous and categorical. variables, respectively. Abbreviations: BMI, body mass index; SSB, sugar-sweetened beverages.

**Table 2 nutrients-13-04563-t002:** Association between SSB consumption with executive function (*n* = 6387).

Executive Function	Estimates (95% Confidence Interval)		
	Crude Model	Model 1 ^a^	Model 2 ^b^
Inhibit			
0 time/week	0 (Reference)	0 (Reference)	0 (Reference)
1 time/week	1.03 (0.42, 1.64) ^c^	0.97 (0.35, 1.59) ^c^	0.87 (0.26, 1.52) ^c^
≥2 times/week	1.76 (1.18, 2.35) ^c^	1.73 (1.12, 2.33) ^c^	1.55 (0.91, 2.19) ^c^
*p* for trend	<0.0001	<0.0001	<0.0001
Shift			
0 time/week	0 (Reference)	0 (Reference)	0 (Reference)
1 time/week	1.24 (0.64, 1.84) ^c^	1.15 (0.53, 1.77) ^c^	1.03 (0.41, 1.66) ^c^
≥2 times/week	2.11 (1.53, 2.68) ^c^	1.97 (1.37, 2.57) ^c^	1.72 (1.08, 2.36) ^c^
*p* for trend	<0.0001	<0.0001	<0.0001
Emotional control			
0 time/week	0 (Reference)	0 (Reference)	0 (Reference)
1 time/week	1.01 (0.40, 1.62) ^c^	0.95 (0.32, 1.59) ^c^	0.82 (0.18, 1.46) ^c^
≥2 times/week	1.80 (1.22, 2.39) ^c^	1.90 (1.29, 2.52) ^c^	1.67 (1.02, 2.32) ^c^
*p* for trend	<0.0001	<0.0001	<0.0001
Initiate			
0 time/week	0 (Reference)	0 (Reference)	0 (Reference)
1 time/week	1.26 (0.66, 1.85) ^c^	1.23 (0.61, 1.84) ^c^	1.10 (0.48, 1.72) ^c^
≥2 times/week	2.49 (1.92, 3.07) ^c^	2.39 (1.79, 2.98) ^c^	2.16 (1.53, 2.80) ^c^
*p* for trend	<0.0001	<0.0001	<0.0001
Working memory			
0 time/week	0 (Reference)	0 (Reference)	0 (Reference)
1 time/week	1.24 (0.63, 1.84) ^c^	1.25 (0.63, 1.87) ^c^	1.10 (0.48, 1.73) ^c^
≥2 times/week	2.25 (1.67, 2.83) ^c^	2.27 (1.67, 2.87) ^c^	2.01 (1.38, 2.64) ^c^
*p* for trend	<0.0001	<0.0001	<0.0001
Plan/organize			
0 time/week	0 (Reference)	0 (Reference)	0 (Reference)
1 time/week	1.33 (0.74, 1.93) ^c^	1.23 (0.62, 1.84) ^c^	1.12 (0.51, 1.74) ^c^
≥2 times/week	2.86 (2.29, 3.43) ^c^	2.69 (2.09, 3.28) ^c^	2.47 (1.84, 3.10) ^c^
*p* for trend	<0.0001	<0.0001	<0.0001
Organization of materials			
0 time/week	0 (Reference)	0 (Reference)	0 (Reference)
1 time/week	1.19 (0.60, 1.79) ^c^	1.09 (0.48, 1.71) ^c^	1.00 (0.38, 1.63) ^c^
≥2 times/week	2.60 (2.02, 3.17) ^c^	2.48 (1.88, 3.08) ^c^	2.27 (1.64, 2.90) ^c^
*p* for trend	<0.0001	<0.0001	<0.0001
Monitor			
0 time/week	0 (Reference)	0 (Reference)	0 (Reference)
1 time/week	1.27 (0.67, 1.86) ^c^	1.17 (0.57, 1.78) ^c^	1.11 (0.50, 1.72) ^c^
≥2 times/week	2.37 (1.80, 2.94) ^c^	2.13 (1.55, 2.72) ^c^	1.99 (1.37, 2.61) ^c^
*p* for trend	<0.0001	<0.0001	<0.0001
BRI			
0 time/week	0 (Reference)	0 (Reference)	0 (Reference)
1 time/week	1.22 (0.61, 1.83) ^c^	1.15 (0.52, 1.78) ^c^	1.02 (0.39, 1.66) ^c^
≥2 times/week	2.13 (1.54, 2.72) ^c^	2.11 (1.50, 2.73) ^c^	1.87 (1.22, 2.51) ^c^
*p* for trend	<0.0001	<0.0001	<0.0001
MI			
0 time/week	0 (Reference)	0 (Reference)	0 (Reference)
1 time/week	1.38 (0.77, 1.99) ^c^	1.31 (0.69, 1.93) ^c^	1.18 (0.55, 1.81) ^c^
≥2 times/week	2.90 (2.31, 3.49) ^c^	2.75 (2.14, 3.35) ^c^	2.50 (1.86, 3.14) ^c^
*p* for trend	<0.0001	<0.0001	<0.0001
GEC			
0 time/week	0 (Reference)	0 (Reference)	0 (Reference)
1 time/week	1.42 (0.80, 2.04) ^c^	1.37 (0.74, 2.00) ^c^	1.22 (0.59, 1.86) ^c^
≥2 times/week	2.85 (2.25, 3.45) ^c^	2.72 (2.10, 3.34) ^c^	2.44 (1.79, 3.09) ^c^
*p* for trend	<0.0001	<0.0001	<0.0001

Abbreviations: BRI, behavioral regulation index; GEC: global executive composite; HI: hyperactivity index; MI, metacognition; SSB, sugar-sweetened beverages. ^a^ Adjusted for sex, age, siblings, monthly household income, parental education, parental smoking status, outdoor exercise and body mass index. ^b^ Additionally adjusted for fried food, fish or fish products, and milk, or dairy-products. ^c^ Statistically significant association (*p*-value < 0.05).

**Table 3 nutrients-13-04563-t003:** Association between SSB consumption with executive dysfunction (*n* = 6387).

Executive Dysfunction	Odds Ratio (95% Confidence Interval)		
	Crude Model	Model 1 ^a^	Model 2 ^b^
Elevated BRI			
0 time/week	1.00 (Reference)	1.00 (Reference)	1.00 (Reference)
1 time/week	1.25 (1.05, 1.48) ^c^	1.27 (1.05, 1.54) ^c^	1.23 (1.02, 1.50) ^c^
≥2 times/week	1.54 (1.31, 1.81) ^c^	1.55 (1.29, 1.86) ^c^	1.45 (1.19, 1.76) ^c^
*p* for trend	<0.0001	<0.0001	0.0002
Elevated MI			
0 time/week	1.00 (Reference)	1.00 (Reference)	1.00 (Reference)
1 time/week	1.21 (1.02, 1.44) ^c^	1.22 (1.01, 1.48) ^c^	1.21 (1.00, 1.47) ^c^
≥2 times/week	1.74 (1.49, 2.04) ^c^	1.72 (1.44, 2.05) ^c^	1.70 (1.41, 2.05) ^c^
*p* for trend	<0.0001	<0.0001	<0.0001
Elevated GEC			
0 time/week	1.00 (Reference)	1.00 (Reference)	1.00 (Reference)
1 time/week	1.13 (0.94, 1.35)	1.15 (0.95, 1.40)	1.14 (0.94, 1.39)
≥2 times/week	1.68 (1.42, 1.98) ^c^	1.67 (1.40, 2.00) ^c^	1.62 (1.34, 1.96) ^c^
*p* for trend	<0.0001	<0.0001	<0.0001

Abbreviations: BRI, behavioral regulation index; GEC: global executive composite; MI, metacognition; SSB, sugar-sweetened beverages. ^a^ Adjusted for sex, age, siblings, monthly household income, parental education, parental smoking status, outdoor exercise and body mass index. ^b^ Additionally adjusted for fried food, fish or fish products, and milk, or dairy-products. ^c^ Statistically significant association (*p* value < 0.05).

## Data Availability

The data presented in this study are available on request from the corresponding author. The data are not publicly available since the participants of this study did not agree for their data to be shared publicly.

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
