# Peer review of "Association between Sugar-Sweetened Beverage Consumption and Executive Function in Children"

_nutrients, 2021, doi:10.3390/nu13124563_

Round 1
Reviewer 1 Report
Manuscript Number: nutrients-1455557
Title: Association between sugar-sweetened beverage consumption with executive function in children
The submitted manuscript focusses on the association of SSB and executive function. The study design and randomization is presented understandable.
Minor comments:
However, the SSB that were investigated in the article included also orange juice that might be a healthier option that the other mentioned beverages. Is there a possibility to separately analyse the effect of the orange juice? Further it would be interesting to compare not only more that 2 servings per week, rather more that 5 (for example) servings would be interesting? Is there also an association? Is it stronger? It would be interesting to know.
Author Response
Title: Association between sugar-sweetened beverage consumption with executive function in children
The submitted manuscript focusses on the association of SSB and executive function. The study design and randomization is presented understandable.
Response: Thank you very much for the positive feedback on our manuscript and for providing us with constructive suggestions. As suggested, we have revised our manuscript to address all of the suggestions as detailed below.
Point 1:
Minor comments:
However, the SSB that were investigated in the article included also orange juice that might be a healthier option that the other mentioned beverages. Is there a possibility to separately analyse the effect of the orange juice? Further it would be interesting to compare not only more that 2 servings per week, rather more that 5 (for example) servings would be interesting? Is there also an association? Is it stronger? It would be interesting to know.
Response: We appreciate this constructive and valuable comment. Orange juice consumption assessed in this study was not 100% fruit juice that could provide essential vitamins and minerals, but is fruit drinks that do not contain very much fruit juice and do not contain vitamins and minerals. Orange juice drinks, such as Minute Maid, were very popular in China. Therefore, the orange juice drink, a sweetened beverage, was hypothesized to be unhealthy for human health. To clarify this to readers, we replace “orange juice” with “fruit drinks (like orange juice drink etc.,)”.
The distribution of SSB consumption status was highly skewed, and transformation of data was not feasible owing to the large number of people who reported never drinking SSB and who drunk ≥2 times/week. Therefore, the frequency of SSB consumption was aggregated and then a new intake category was categorized in order to ensure an adequate number of participants in each group. SSB consumption was examined as a 3-level variable: 0 time/week, 1 time/week, and ≥2 times/week. In analyses, we restricted to children who reported drinking SSB to assess the dose-response relationship between the total servings of SSB consumption a week (as a continuous variable) and executive function. And we found that per 1 serving increase in SSB consumption was associated with all scores of BRIEF, with the estimates ranging from 0.21 (95%CI: 0.03, 0.40) to 0.35 (95%CI: 0.16, 0.54) (Table A2).
Reviewer 2 Report
The study deals with the association between sugar-sweetened beverages (SSB) consumption and executive function among school aged children. The problem is important for children health but not sufficiently studied as yet. However, the existing data could allow provide more information on this subject in the Introduction. The data on the mechanisms of the influence of sweet beverages obtained in animal studies should be transferred from the Discussion to the Introduction. The effects of the consumption of sweets in general and other nutrients on the executive function in children should be presented – in Introduction or Discussion. As regards the scope of the research, it is difficult to understand why the consumption of other sources of sugar was not assessed in this study? The dietary assessement (fried foods, fish and milk products) performed in the investigations might characterize the correctness of nutrition, but it does not contribute to broadening the knowledge about the issue studied.
Author Response
The study deals with the association between sugar-sweetened beverages (SSB) consumption and executive function among school aged children. The problem is important for children health but not sufficiently studied as yet. However, the existing data could allow provide more information on this subject in the Introduction. The data on the mechanisms of the influence of sweet beverages obtained in animal studies should be transferred from the Discussion to the Introduction. The effects of the consumption of sweets in general and other nutrients on the executive function in children should be presented – in Introduction or Discussion. As regards the scope of the research, it is difficult to understand why the consumption of other sources of sugar was not assessed in this study? The dietary assessement (fried foods, fish and milk products) performed in the investigations might characterize the correctness of nutrition, but it does not contribute to broadening the knowledge about the issue studied.
Response: We appreciate this constructive and valuable comment. As suggested, we have added texts to described the mechanisms of the influence of sweet beverages on brain health “Animal studies have shown that sugar could induce the increases in mediators of inflammation (e.g., IL-6 and IL-1β) and oxidative stress as well as decrease in neurotrophins, and these intermediate factors subsequently altered brain structure and function (Hsu et al., Hippocampus 2015; Sanguesa et al., Mol. Neurobiol. 2018; Agrawal et al., J. Cereb. Blood Flow Metab. 2016)” (Pages 1-2 Lines 48-50). Owing to the limited scope of research (the association between SSB and executive function) and the scarce of studies investigating the association between the sugar in other sources, we did not describe the study situation regarding the effects of other nutrients or sugar in general. Studies in the future should be conducted to examine the multi-nutrients on executive function.
In our study, we did not collect the information on other sources of sugar, such as 100% fruit juice and sugar sweetened snacks, and the lack of information was recognized as a limitation in our study. To clarify this point to readers, we have added “We did not measure dietary intake, and therefore intake of foods high in sugar could not be accounted for.” in the Discussion (Page Lines 28-30).
References:
Hsu, T.M.; Konanur, V.R.; Taing, L.; Usui, R.; Kayser, B.D.; Goran, M.I.; et al. Effects of sucrose and high fructose corn syrup consumption on spatial memory function and hippocampal neuroinflammation in adolescent rats. Hippocampus 2015,25,227-239. https://doi.org/10.1002/hipo.22368.
Sanguesa, G.; Cascales, M.; Grinan, C.; Sanchez, R.M.; Roglans, N.; Pallas, M.; et al. Impairment of novel object recognition memory and brain insulin signaling in fructose- but not glucose-drinking female rats. Mol. Neurobiol. 2018,55,6984-6999. https://doi.org/10.1007/s12035-017-0863-1.
Agrawal, R.; Noble, E.; Vergnes, L.; Ying, Z.; Reue, K.; Gomez-Pinilla, F. Dietary fructose aggravates the pathobiology of traumatic brain injury by influencing energy homeostasis and plasticity. J. Cereb. Blood Flow Metab. 2016,36:941-953. https://doi.org/10.1177/0271678X15606719.